# Dynamic Evaluation of Natural Killer Cells Subpopulations in COVID-19 Patients

**DOI:** 10.3390/ijms231911875

**Published:** 2022-10-06

**Authors:** Adina Huțanu, Doina Manu, Manuela Rozalia Gabor, Anca Meda Văsieșiu, Akos Vince Andrejkovits, Minodora Dobreanu

**Affiliations:** 1Department of Laboratory Medicine, George Emil Palade University of Medicine, Pharmacy, Science, and Technology of Targu Mures, 540142 Targu Mures, Romania; 2Department of Laboratory Medicine, Emergency Clinical County Hospital Targu Mures, 540136 Targu Mures, Romania; 3Center for Advanced Medical and Pharmaceutical Research, George Emil Palade University of Medicine, Pharmacy, Science, and Technology of Targu Mures, 540136 Targu Mures, Romania; 4Department of Economic Science, Faculty of Economics and Law, George Emil Palade University of Medicine, Pharmacy, Science, and Technology of Targu Mures, 540136 Targu Mures, Romania; 5Department of Infectious Diseases, George Emil Palade University of Medicine, Pharmacy, Science, and Technology of Targu Mures, 540142 Targu Mures, Romania

**Keywords:** COVID-19, SARS-CoV-2, natural killer cells, PD-1

## Abstract

The aim of the study was to evaluate the dynamic changes of the total Natural Killer (NK) cells and different NK subpopulations according to their differentiated expression of CD16/CD56 in COVID-19 patients. Blood samples with EDTA were analyzed on day 1 (admission moment), day 5, and day 10 for the NK subtypes. At least 30,000 singlets were collected for each sample and white blood cells were gated in CD45/SSC and CD16/CD56 dot plots of fresh human blood. From the lymphocyte singlets, the NK cells subpopulations were analyzed based on the differentiated expression of surface markers and classified as follows: CD16^-^CD56^+/++^/CD16^+^CD56^++^/CD16^+^CD56^+^/CD16^++^CD56^−^. By examining the CD56 versus CD16 flow cytometry dot plots, we found four distinct NK sub-populations. These NK subtypes correspond to different NK phenotypes from secretory to cytolytic ones. There was no difference between total NK percentage of different disease forms. However, the total numbers decreased significantly both in survivors and non-survivors. Additionally, for the CD16^-^CD56^+/++^ phenotype, we observed different patterns, gradually decreasing in survivors and gradually increasing in those with fatal outcomes. Despite no difference in the proportion of the CD16^−^CD56^++^ NK cells in survivors vs. non–survivors, the main cytokine producers gradually decline during the study period in the survival group, underling the importance of adequate IFN production during the early stage of SARS-CoV-2 infection. Persistency in the circulation of CD56^++^ NK cells may have prognostic value in patients, with a fatal outcome. Total NK cells and the CD16^+^CD56^+^ NK subtypes exhibit significant decreasing trends across the moments for both survivors and non-survivors.

## 1. Introduction

The clinical evolution of COVID-19 is greatly dependent on the primary antiviral immune response of the host. Both an excessive or diminished antiviral response are involved in the pathology of severe disease. A subdued antiviral response will allow the virus to replicate without control, while an exacerbated response involving the complement activation and cytokine storm will eventually lead to multiorgan destruction and dysfunction. Thus, it is important to characterize the cells that are involved during the early stages of the infection. 

Leucocytes are the most important cells involved in the primary immune response, each of the cell classes being specialized in distinct mechanisms of protection. NK cells are important and are acting at the interface between innate and adaptive immune responses. They interact with viruses and tumor cells without prior sensibilization through direct cytotoxicity, antibody-dependent cellular cytotoxicity (ADCC), and cytokine production [1]. In addition, NK cells seem to have a part in the “innate immune memory” by way of epigenetic reprogramming of transcription factors that leads to metabolic changes. Some of these factors, such as H3 histone lysin 4 monomethylation or Set7 methyltransferase, were reviewed in the context of trained immunity [2]. 

NK cells express several markers on their surface with different important functions in the modulation of their activity, such as attaching to the target cell, linking IgG, and preventing unnecessary activation. All these surface markers, clusters of differentiation (CDs) and activating/inhibitory receptors have significance in NK activity regulation. The most representative markers for NK cells are CD56, an adhesion molecule that allows the attachment to the target cell, and CD16, the receptor for FcγRIIIA, important in intermediating ADCC [3]. CD107a, a lysosomal-associated membrane protein, is expressed on the NK cell membrane after Major Histocompatibility Complex (MHC) stimulation and reflects cytotoxic activity and cytokine secretion [4], while CD57 is a marker of cell maturation expressed together with Killer Immunoglobulin-like Receptors (KIR) [5]. 

In addition to the above-mentioned CDs, the mature, functional, and fully educated NK cells express CD226, an adhesion molecule involved in the target recognition, and KIR, which is important in regulating NK cells activity, tailored in accordance to the host’s MHC [5,6].

Several subtypes of NK cells emerged after analyses of peripheral blood mononuclear cells (PBMC) in healthy subjects, as well as in those infected with Human Immunodeficiency Virus (HIV) [7] or SARS-CoV-2 viruses [8]. 

In humans, the NK population is classified into two predominant classes, mainly based on the intensity of expression of CD56 and CD16. Thus, the potent cytolytic cells express CD16 (named CD16+ or CD16 low) and with lower intensity CD56, and represent the major population in the periphery (90% of NK cells) [9]. The cells with secretory phenotype, mainly located in the secondary lymphoid tissue (SLT) [1] with a minor fraction below 10% in peripheral blood, express CD56^bright^ (or CD56^++^). These cells are considered as immature precursors of the fully differentiated NK cells [9]. 

In addition to the aforementioned NK types, the literature describes a few additional subtypes of NK cells based on the differentiated expression of CD16 and CD56. In this regard, in our study we will take into consideration four types of NK cells, with focus on the main NK population:✓CD16^-^CD56^+/++^ with poor cytotoxic, but high cytokine production capacity, having a secretory phenotype.✓CD16^+^CD56^++^ represents a NK subset with both secretory and cytolytic properties. These cells were studied in the context of melanoma and researchers found differences between CD56^bright^ CD16^-^ and/or CD16^+^ NK cells with higher activation state for the CD16^+^ cells, higher degranulation capacity, and higher cytokine production [9,10]. However, in the melanoma study, this cell type was only found in a metastatic lymph node [9]. IFNγ producing circulating CD56^++^ NK cells were found by de Jonge et al. in melanoma patients inversely correlated with survival [11].✓CD16^+^CD56^+^ cells have a potent cytolytic activity and represent the majority of the circulating NK cells [1,12].✓CD16^++^CD56^-^ is another subtype of NK cells with a potential role in different types of infections, especially in HIV-infected patients [12]. Forconi et al. considered this kind of NK cells as an adaptation model of CD16^+^CD56^+^ in chronic infections, mainly focused on ADCC, as CD16 is highly expressed [13].

Milush et al. have particularly studied the CD16^++^CD56^−^ subset of NK cells by using CD7 as an additional marker for NK cells in HIV infection. They found that cells with the phenotype CD3^−^ CD7^+^CD16^++^CD56^−^ are also mature NK cells, expressing KIRs and NKG2A^+^, which in HIV infection respond with reduced secretion of Interferon γ (IFNγ) [14]. NKG2A^+^ is a marker expressed earlier during the education and maturation process and is gradually lost [15]. Additionally, the expressions of granzyme B and perforin were lower for CD7^+^CD16^++^CD56^−^ cells compared to CD16^+^CD56^+^ cells. Additionally, the perforin and granzyme B production was lower in HIV-infected patients compared to controls, generating the hypothesis that during viral infection (including HIV infection), these CD16^++^CD56 cells are mature NK cells that have recently encountered a target cell in vivo, as it is known that NK cells are multiple hit cells, thus gradually losing their enzymatic equipment [14].

One important regulatory protein found on the lymphocyte membrane is Programmed cell death protein 1 (PD-1), which is found mainly on T and B lymphocytes, and to a lesser extent on monocytes, dendritic cells, and NK cells [16]. Along with its ligand (PD-L1), its major role is to act as an immune checkpoint, down-regulating the immune response and favoring the self-tolerance by modulating the apoptosis of antigen-specific T cells and regulatory T cells [17]. An increased PD-1 expression on cell membrane during the viral infections will eventually lead to cells’ exhaustion, apoptosis, and impaired immune response. As reviewed by Aghbash et al., during SARS-CoV-2 infection, the lymphopenia is accompanied by increased PD-1 expression in both CD8^+^ and CD4^+^ T cells [17]. 

Cancer research revealed that the antitumoral activity of NK cells is related to the differential expression of PD-1 molecules [18]. The interaction between PD-1 with its ligand PD-Ll exerts many effects by inhibiting the Ly T cytokines productions and the T cells proliferation, overall engaging a down-regulated signal for immune cell activities [19]. For NK cells, a higher PD-1 expression implies a reduced IFN gamma secretion, a lower degranulation degree, and reduced antitumoral properties [18]. As for the SARS-CoV-2 infection, the increased expression for PD-1 and its ligand is related to disease severity and exhaustion of the immune cells, but also could also serve as a potential therapeutic approach by inhibiting the immune checkpoints [20].

Given the high variety of the NK cell subpopulations found in peripheral blood and considering their paramount role during an early antiviral immune response, we intended to characterize the NK cell subtypes in patients infected with SARS-CoV-2. The first aim of our study was to assess the dynamic changes of NK cell subsets and their PD-1 expression through analysis of the peripheral blood lymphocytes by flow cytometry, in different moments of disease evolution. The second goal of our study was to correlate the NK subpopulation profile with the disease severity and outcome.

## 2. Results

### 2.1. Characterization of the Patients and Baseline Results

The study on the NK cells populations was performed at three different time points (Day 1-5-10: MS_0, MS_5, and MS_10) for analysis of the dynamic changes in NK cells profile and the possible correlation with patients’ status and outcomes were investigated. A total of 53 patients, male *n* = 27 (50.9%) and female *n* = 26 (49.1%), were recruited (*p* > 0.5). For the MS-0 assessment, *n* = 6 (11.3%) had mild disease, *n* = 14 (26.4%) had moderate forms, and *n* = 33 (62.3%) had severe/critical COVID-19 forms. For the second time point, only 49 patients remained for assessment, while for the third time point, only 20 were analyzed regarding the NK cells subpopulation. A total of 42 patients survived (79.2%), while *n* = 11 (20.8%) died before the study ended (4 of them before 5th day from admission). The demographic characteristics and clinical status of the patients on admission (M_0) are provided in detail in Table 1.

No notable difference was found for MS_0, MS_5, and MS_10 between survivors and non-survivors for total NK (% and absolute number) and NK subsets. 

### 2.2. Dynamic Analysis of NK Cell Subpopulations 

Based on the expression of surface markers, the NK singlets were further analyzed for dynamic changes; in this regard, lymphocyte singlets were classified based on the different expressions of CD16/CD56 markers. The dynamic evolution of total NK (%) and for the CD16^−^CD56^+^/56^++^, CD16^+^CD56^++^, CD16^+^CD56^+^, and CD16^++^CD56^−^ subsets is shown in Figure 1.

In the fatal outcome group, for between moments comparison, there was a significant difference for total NK (%) with lower NK cells, a median of 10.90% (5.80–17.30) on day 10th compared to the admission moment, with a median of 16.00% (7.60–49.20), *p* = 0.041. For the NK subpopulations (%), we found a significant difference between moments only for CD16^+^CD56^+^, which otherwise represent the most abundant cell population; we found a significant difference between the time points. The mature NK cells CD16^+^CD56^+^ were significantly higher on admission compared to MS_10, with a median of 13.00% (6.00–46.40) vs. 5.50% (3.60–15.70), *p* = 0.016. For the rest of the NK subtypes, CD16^-^CD56^+/++^, CD16^+^CD56^++^, and CD16^++^CD56^-^ respectively, there was no significant difference between moments.

For the survival group, we found a significant difference for total NKs and CD16^+^CD56^++^ and CD16^+^CD56^+^ subpopulations between MS_0 and MS_10. Values were considerably higher at moment MS_0 and decreased substantially at MS_10. For total NK cells, the median for MS_0 was 18.50% (2.60–42.40), significantly higher compared to MS_10 with a median of 9.60% (3.90–20.70), *p* = 0.0001. The median values recorded for CD16^+^CD56^++^ were 0.10% (0.00–0.70) for admission moment vs. 0.00% (0.00–0.20) on MS_10, *p* = 0.024. The most prevalent NK subpopulation, CD16^+^CD56^+^ constantly decreased from the first moment, a median of 14.00% (1.40–39.30) compared to the last moment of analysis when the median was 6.60% (2.90–10.30), *p* = 0.0008. All the serial changes between moments for NK subpopulations aspects are detailed in Figure 1. The analysis of the difference between severity groups, for overall patients (survivors and non-survivors) for the MS-0 (first day on admission), is detailed in Table 2; all non-surviving subjects were in the critical/severe group.

We noticed that the CD16^-^CD56^+/++^ population expresses opposite evolution during the disease course, from MS_0 to MS_10, gradually decreasing in survivors and increasing in non-survivors, especially visible for the highest values of the intervals (*p* = NS). In survivors, we observed a gradual reduction in the number of CD16^-^CD56^+/++^ subtypes, especially from the first to the second moment (acute phase of the viral disease); in non-survivors, these cell subtypes plateaued, with a slight increase in the third moment.

As can be seen in Table 2, in patients with mild disease forms at admission, total NK cells are significantly lower compared to moderate and critical forms (*p* = 0.032 and 0.029, respectively). The same pattern with a lower median in mild cases compared with a higher median in severe patients is observed for the CD16^+^CD56^+^ phenotype.

### 2.3. Expression of PD-1 on NK Cells according to Disease Severity and Outcome

The PD-1 expression estimated by median fluorescence intensity (MFI) on natural killer cells was analyzed in accordance with disease severity. PD-1 and its ligand are important markers of modulation of immune cells and important markers of cell exhaustion, including of the NK cells. The study population was dichotomized into two severity subgroups, due to the reduced number of patients with mild disease forms. Thus, mild + moderate forms were compared with critical forms of disease regarding the expression of PD-1 molecules, and no difference was found. Similarly, when the MFI of PD-1 was observed in survivors vs. non-survivors, no significant difference was found (data not shown). 

## 3. Discussion

The aim of our study was to analyze the dynamic evolution of the total NK cells and of different phenotypes, as they were identified on the flow cytometry plots according to the expression of CD16/CD56. For this purpose, lymphocytes were analyzed on the 1st day, 5th day, and 10th day after admission. 

The NK cells are important cells of innate immune response with a paramount role in early antiviral defense. The inadequate NK reaction during SARS-CoV-2 infection will lead to an unbalanced antiviral response by increasing the pathological over the protective response. As we reviewed earlier, after the virus sensing, a proper endogenous IFN synthesis is the key to an efficient antiviral response [21]. Both the inhibitory effect of the SARS-CoV-2 on IFN type I and III synthesis and a delayed IFN production will lead to an increased viral replication and will impede viral load decline, all contributing to more severe COVID-19 forms. This is explained by the SASR-CoV-2-encoded proteins which interfere with viral recognition and IFN signaling pathways [22]. After virus recognition, the NK cells undergo activation and phenotype differentiation. The main NK cytokine producers, CD16^neg^CD56^high^, will increase in number and *IFN* gene up-regulation. Apart from the phenotypes of NK cell profile alteration, the NK exhaustion highlighted by PD-1 increased expression also characterizes the severe disease due to the hyper-inflammatory status [23]. In our study, in the fatal outcome group, there were elderly patients with severe disease, which is not surprising, since in this age group, both innate (through dysfunctional NK and impaired IFN synthesis) and adaptive immune systems undergo immunosenescence.

The major findings in our study are related to the frequency of the CD16^−^CD56^++^ subtypes of NK cells, exhibiting an opposite trend, decreasing gradually in survivors and increasing gradually in non-survivors. Their main activity is cytokine production and cytolytic activity to a lesser extent or not at all. The CD56^++^ NK cells are the main producers of cytokines, having an important role in modulating the immune response [12]. More importantly, the hallmark of their cytokine production is INF gamma a cytokine important in tailoring the early viral immune response [23].

In a study of single-cell immune profiling, researchers underline the difference between early-stage asymptomatic vs. moderate/severe SARS-CoV-2 infected patients. While the main cytokine producers CD16^neg^CD56^high^ NK cells were found in higher proportion in asymptomatic patients with the high expression of *IFNG,* the CD16^+^CD56^dim^ was significantly higher in patients with severe COVID-19. Additionally, there was a specific trend for up-regulation of *IFNG* in accordance to disease stage, with the highest expression at 10 days after symptoms onset [24]. Our results exhibit different patterns regarding the frequency of CD16^neg^CD56^high^ (named CD16^−^CD56^+/++^) in relation to disease severity, higher in moderate and severe vs. mild disease, though without statistical significance, probably due to the small sample size of our study. However, these observations are for the MS_0 time point, the time of the patients’ admission to the hospital (MS_0).

Although we did not find any difference in the proportion of the CD16^−^CD56^++^ NK cells in the survivors vs. non–survivors, in the survivors’ group, the main cytokine producers were higher on admission compared to the second and third analyzing moments, gradually declining thereafter from a median of 2.70% at MS_0 to a median of 1.80% for the MS_5 (*p* = 0.094) and a median of 1.50% for MS_10. This could be translated into the fact that an adequate IFN production during the early stage of the SARS-CoV-2 infection promotes the reduction of viral load and replication, with consecutive favorable evolution [25]. On the contrary, within the non-survivor group, these NK subsets were found in lower proportion during the earliest stage of the infection and in higher proportion on day 10 (*p* = NS). This is in line with the unfavorable outcome, as it is well known that the IFN synthesis during the late viral infection enhances the inflammatory response and cytokine storm [23,26]. 

There was no difference between survivors and non-survivors regarding the absolute count of NK or percent of cytolytic cells (CD16^+^CD56^+^ and CD16^++^CD56^−^ subtypes) for the first assessed moment. However, in all patients regardless of the fatal disease outcome, there was a significant gradual decrease in CD16^+^CD56^+^ cells from moment MS_0 to MS_10, *p* = 0.0008 for survivors and *p* = 0.016 for non-survivors, while the phenotype CD16^++^CD56^-^ remains unchanged during the evaluation. Wang et al. dynamically analyzed the evolution of the immune cells in COVID-19 patients and found that in those with fatal outcome, the NK cells gradually declined, while in survivors, NK cells increased from the onset to the late stage of the disease [27]. A recent study that analyzed the NK subtypes reported a significantly lower frequency of CD16^high^CD56 ^dim^ NK subsets with subsequently higher expression of inhibitory receptors in COVID-19 patients compared to naïve and convalescent subjects, with no difference in CD56^high^ NK cells [8].

In shaping the immune response, not only the frequency of NK cells is important but also the cells’ functions. During SARS-CoV-2 infection, the degranulation potency of NK cells was analyzed by Garcinuño et al., finding an impaired degranulation capacity in severe COVID-19 patients compared to healthy and asymptomatic subjects [28].

As Deng et al. reviewed, the number and function of NK cells are related to COVID-19 evolution, since these features are affected more frequently in severe SARS-CoV-2 infection. A reduced number of NK cells is associated with a lower decline in viral load rate, weaker antibody response, and lower survival rate among SARS-CoV-2-infected subjects [29]. 

Despite no difference being found in the frequency of total NK cells between survivors and non-survivors, in our study, the different distribution of the NK subtypes across the different disease forms or between survivors and non-survivors could provide an informative insight into the pathological alteration during the SARS-CoV-2 infection. Both in survivors and non-survivors, we found a significant decline in total NK % across the study moments with the highest NK values on admission and the lowest at the end of the study protocol (10th day). The advantage of our study is that it measures dynamically the evolution of different NK phenotypes 10 days from admission.

A very recent study performed by Niedzwiedzka-Rystwej et al. investigated the dynamic evolution of T, B, and NK cells during 14 days of hospitalization, both ICU and non-ICU, and the role of PD-1/PD-L1 as a biomarker for COVID-19 severity in hospitalized patients. They found that for the admission moment, the expression of PD-1/PD-L1 was higher for CD4^+^ and CD19^+^ cells, but not for CD8^+^ cells, and did not change during the study period [16], although the study did not provide the PD-1 expression on NK cells. In our study, the PD-1 expression was evenly distributed across the different time moments and disease outcomes for all NK subpopulations. The same authors reported no difference in NK percentage between ICU and non-ICU patients, for all three time points, but NK cells were significantly higher in healthy controls compared to COVID-19 patients. Additionally, the NK population was found significantly higher in patients with fatal outcomes compared to survivors, and the percentage did not differ significantly across the moments [16]. In our study, the total NK % was slightly higher in survivors (not statistically significant), and for both fatal and non-fatal outcomes, the dynamic changes across the moments follow the same trend: higher NK percentage on admission with subsequent reduction of percentage during the follow-up period. Regarding the most important NK phenotypes, secretory and cytolytic, there was a specific pattern. While for the secretory NK subtypes in survivors, the number decreased gradually during the study period, in non-survivors, there was an opposite trend, with higher values for the third moment compared to admission evaluation. This trend in survivors could be in line with the important role of IFN secretion during the early stages of the disease for rapid viral clearance, while in patients with fatal outcomes, maintaining an increased number of secretory NKs will enhance the inflammatory response with detrimental results. For the main NK periphery cells displaying CD16^+^CD56^+^ markers, both for favorable and unfavorable outcomes, the percentage decreased significantly across the study period. 

The study has some limitations, including the small sample size and the inconsistency of the NK subtype analysis at all moments for all recruited subjects. Additionally, we did not have a healthy control group for comparison purposes and the cytokines levels were not assessed in these patients for a further correlation with the NK activity. An extended study that intends to analyze cells of innate and adaptive immune response along with their expression of PD-1 represents the next step in the characterization of the immune response in patients with SARS-CoV-2 infection.

## 4. Materials and Methods

### 4.1. Patient Recruitment and Sample Collection

Blood samples from 56 recruited patients infected with SARS-CoV-2 and admitted to the 1st Infectious Disease Clinic of Targu Mures between December 2021 and February 2022 were assessed at three time points for the dynamic analysis of a lymphocyte subpopulation during disease evolution. Of them, two patients were excluded as technical outliers and one for a negative RT-PCR test result. Blood samples were collected in Ethylenediaminetetraacetic Acid (EDTA) tubes on the 1st day (admission moment), defined as MS_0, 5th day, defined as MS_5, and 10th day, defined as MS_10. Collected samples were immediately sent to the laboratory and analyzed by flow cytometry protocol. Patients were designated with mild, moderate, and severe/critical COVID-19 disease, in accordance with clinical status and criteria according to the WHO Interim Guidance on Clinical Management of COVID-19.

Definition of disease severity was as follows:
-Mild grade (Stage I) was defined as a disease with few symptoms (low fever, cough, fatigue, anorexia, shortness of breath, myalgias), without evidence of viral pneumonia or hypoxia.-Moderate grade (Stage II) was defined as a disease with fever and respiratory symptoms, associated with pulmonary imaging findings, but no signs of severe pneumonia, including SpO2 ≥ 90% on room air.-Severe grade (Stage III) was defined as a disease with severe pneumonia, with clinical signs of pneumonia (fever, cough, dyspnea) plus one of the following: respiratory rate > 30 breaths/min; severe respiratory distress; or SpO2 < 90% on room air.-Critical grade (Stage IV) was defined as acute respiratory distress syndrome (ARDS), septic shock, and/or multiple organ dysfunction.


The patients who were discharged from the hospital were designated as survivors, while those who died during hospitalization were non-survivors.

The study was approved by the Ethics Committee of the George Emil Palade University of Medicine, Pharmacy, Science, and Technology of Targu Mures (No.1237/08.01.2021) and by the Institutional Ethics Committee of Clinical County Hospital Targu Mures (No.19038/21.12.2020). The study was conducted in accordance with the Helsinki Declaration, and all participants signed the informed consent prior to admission to the study.

Inclusion criteria: SARS-CoV-2 infected patients, confirmed by real-time polymerase chain reaction (RT-PCR), who signed the informed consent for the recruitment into the study. Exclusion criteria: age < 18 years, immunocompromised patients (HIV/AIDS; cancer, transplant, pregnancy, or other similar pathologies), and patients with incomplete data. 

### 4.2. Natural Killer Lymphocyte Subset Analysis

Two EDTA tubes were collected for each patient: one EDTA tube was used for complete blood count using Sysmex XS-800i hematology analyzer, while the second sample of peripheral blood was used for immunophenotyping using the BD FACSAria III flow cytometer.

For the analysis of NK cells by flow cytometry, mouse antibodies with specific reactivity against human antigens were used, details regarding antibodies characteristics and cytometer configuration are presented in Table 3.

The detailed protocol for NK cell assessment is described below. A volume of 50 μL whole blood was incubated with the mixture of antibodies, previously prepared in staining buffer (BD cat. no. 554656), in optimal concentrations, depending on the number of cells subjected to fluorescent labeling with specific antibodies. After 15 min of incubation at room temperature in the dark, the samples were treated for red blood cell lysis with BD Pharm Lyse ™ lysing solution (BD cat. no. 555899), followed by a washing step with PBS. The leucocytes were further analyzed using the BD FACSAria III cytometer, data were acquired and analyzed with the BD FACSDiva^TM^ program version 8.0.1. For each working section, the correct operation of the cytometer was checked, using BD setup and tracking beads (BD cat. no. 655050); no significant variations in the acquisition parameters were found between runs. 

During the flow cytometry acquisition, at least 30,000 singlets were collected for each human blood sample, and white blood cells were gated in CD45/SSC and CD16 vs. CD56 dot plots of fresh human blood. From the lymphocyte singlets, the NK cells subpopulations were analyzed based on the differentiated expression of surface markers CD16 and CD56 and classified from left to right side of Figure 2 as follows: CD16^-^CD56^+/++^/CD16^+^CD56^++^/CD16^+^CD56^+^/CD16^++^CD56^−^. Additionally, the expression of PD-1 surface marker expressed as median fluorescence intensity (MFI) was studied for each NK cell type on monoparametric histograms.

### 4.3. Statistical Analysis

Descriptive statistics was used for all study variables and presented as median (min–max) for variables with non-normal distribution and mean ± SD for variables with normal distribution. For categorical variables, absolute and relative frequencies were used. The normality of distribution for the continuous variable was tested with one Sample Kolmogorov–Smirnov test. An adequate statistical test was used for comparisons between survivors or non-survivors *p* < 0.05 was considered significant. A graphical representation of box plots was used for cell populations. The SPSS 23.0 license was used.

## 5. Conclusions

Despite no difference in the proportion of the CD16^−^CD56^++^ NK cells in survivors vs. non-survivors, the main cytokine producers gradually decline during the study period in the survival group. This is in line with adequate IFN production during the early stage of the SARS-CoV-2 infection, facilitating the control of viral replication and a favorable outcome. Additionally, total NK cells and the CD16^+^CD56^+^ NK subtypes exhibit a significant decreasing trend across the moments. No differences were observed for PD-1 expression on NK subtypes for both survivors and non-survivors. Further studies on the NK cells evolution in the context of different variants or viral strains infections will clarify their importance in COVID-19 outcomes. 

## Figures and Tables

**Figure 1 ijms-23-11875-f001:**
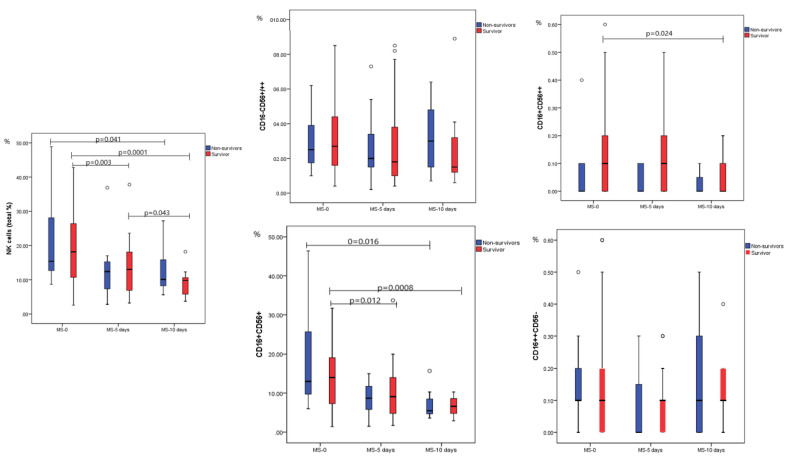
Time point comparison of total NK and NK subpopulations (%) in both survival and non-survival groups. MS = moment of sampling, NK = natural killer, CD = cluster of differentiation; results were obtained after analysis with Mann–Whitney. For the survivors’ group: *n* = 42 for MS_0, *n* = 38 for MS_5, and *n* = 13 for MS_10, while for the non-survivors’ group: *n* = 11 for MS_0, *n* = 11 for MS_5, and *n* = 7 for MS_10. The box plots/figures without mention of *p* values on the graphics belong to comparisons with no significant difference between moments.

**Figure 2 ijms-23-11875-f002:**
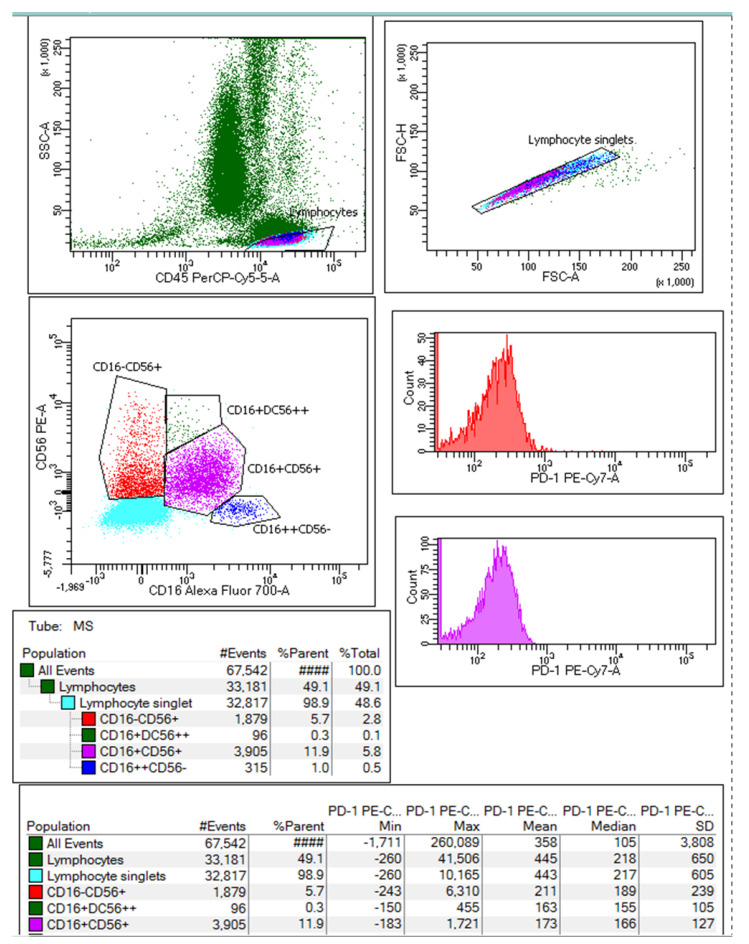
The gating strategy for the NK cells subpopulations was set based on the differentiated expression of surface markers CD16 and CD56. The NK cells population was classified from left to right side of the plot: CD16^−^CD56^+/++/^CD16^+^CD56^++^/CD16^+^CD56^+^/CD16^++^CD56^−^ (image from one patient at MS_0).

**Table 1 ijms-23-11875-t001:** Demographic characteristics of the study group, regarding the disease severity, clinical management, and laboratory parameters in survivors and non-survivors for MS_0.

	Survivors *n* = 42	Non-Survivors *n* = 11	*p*
	**Demographics**		
Age, years ± SD	69.8 ± 13.1	75.6 ± 10.2	0.179
Gender (male), n (%)	22 (52.4%)	5 (45.5%)	0.943
	**Clinical parameters**		
Disease severity			0.014
Mild, n (%)	6 (14.3%)	-	
Moderate, n (%)	14 (33.3%)	-	
Severe/critical, n (%)	22 (52.4%)	11 (100%)	
SaO_2_ %	92.1% ± 5.1	79.6% ± 9.2	<0.0001
Antiviral therapy, n (%)	27 (64.3%)	7 (63.6%)	0.754
Antibiotherapy, n (%)	33 (78.5%)	11 (100%)	0.217
Vaccination	13 (30.9%)	3 (27.3%)	0.894
Comorbidities			
Hypertension	28 (66.7%)	11 (100%)	0.026
Chronic Cardiovascular disease	22 (52.4%)	9 (81.8%)	0.078
Diabetes mellitus	6 (14.3%)	3 (27.3%)	0.307
Asthma	1 (2.4%)	2 (18.2%)	0.044
Chronic Kidney Disease	7 (16.7%)	2 (18.2%)	0.905
Chronic hepatopathy	3 (7.1%)	0 (0%)	0.361
Other	32 (76.7%)	9 (81.8%)	0.691
	**Laboratory parameters**		
Leucocytes (WBC)	8.87 ±4.12	9.42 ±4.14	0.679
Neutrophils (%)	78.68 ± 9.69	84.78 ± 6.14	0.077
Neutrophils (#)	6.83 ± 3.77	8.14 ± 3.97	0.289
Lymphocytes (%)	11.82 (3.33–36.31)	7.31 (4.48–15.49)	0.070
Lymphocytes (#)	0.83 (0.33–2.22)	0.73 (0.30–0.93)	0.050
Monocytes (%)	6.89 (0.68–17.45)	5.48 (3.42–12.68)	0.395
Monocytes (#)	0.56 (0.05–1.84)	0.54 (0.30–0.82)	0.904
	**Natural Killer cells subpopulations**		
NK cells (total %)	18.15 (2.60–42.80)	15.40 (8.70–48.90)	0.684
NK cells (total #)	0.15 (0.02–0.59)	0.13 (0.06–0.30)	0.339
CD16^−^CD56^+/++^	2.70 (0.40–18.30)	2.50 (1.00–6.20)	0.613
CD16^+^CD56^++^	0.10 (0.00–0.70)	0.00 (0.00–0.40)	0.086
CD16^+^CD56^+^	14.0 (1.40–39.30)	13.0 (6.0–46.40)	0.496
CD16^++^CD56^−^	0.10 (0.00–0.60)	0.10 (0.00–0.50)	0.316

Results were obtained using the student *t*-test for parameters with normal distribution, the Mann–Whitney test, for non-parametrical distribution, the chi-squared test for categorical parameters, and *p* < 0.05 for statistics. NK = natural killer cells, SaO_2_ = saturation in oxygen room air; WBC = white blood cells, (%) = percent of cell population, (#) = absolute number of cells. Values are expressed as mean ± SD or median (min-max).

**Table 2 ijms-23-11875-t002:** The analysis of the NK cell subpopulation at admission for all patients, survivors, and non-survivors.

NK cells	Mild (*n* = 6)	Moderate (*n* = 14)	Severe/Critical (*n* = 33)	*p*
NK cells (total %)	10.7 (8.1–19.7)	23.4 (6.8–33.2)	18.7 (2.6–49.2)	0.032 * 0.879 ** 0.029 ***
CD16^-^CD56^+/++^	1.55 (1.20–3.20)	3.55 (0.80–7.70)	2.60 (0.40–18.30)	0.231* 0.753 ** 0.106 ***
CD16^+^CD56^++^	0.10 (0.00–0.20)	0.10 (0.00–0.30)	0.10 (0.00–0.70)	0.496 * 0.421 ** 0.869 ***
CD16^+^CD56^+^	7.25 (4.90–14.40)	18.15 (4.20–31.70)	13.60 (1.40–46.40)	0.069 * 0.762 ** 0.026 ***
CD16^++^CD56^−^	0.25 (0.00–0.60)	0.10 (0.00–0.20)	0.10 (0.00–0.60)	0.291 * 0.817 ** 0.305 ***

Values are expressed in % from total NK cells. NK = natural killer cells, CD = cluster of differentiation; results were obtained after analysis with the Mann–Whitney test, expressed with median and min-max; * = *p* for comparison between mild-moderate, ** = *p* for comparison between moderate-severe, and *** = *p* for comparison between mild-severe/critical disease forms.

**Table 3 ijms-23-11875-t003:** Parameters specifications and BD FACSAria III flow cytometer configuration used for data acquisition during the study protocol.

Excitation LASER	Fluorochrome	Specificity	Relative Brightness	Band-Pass filters (nm)	Mouse Antibody
Blue (488 nm)	BD Pharmingen™ PE	Human CD56	Bright	575/26	BD 555516
	BD Pharmingen™ PE-Cy7™	Human PD-1	Brightest	780/60	BD 561272
	BD Pharmingen™ PerCP-Cy 5.5	Human CD45	Moderate	695/40	BD 567310
Red (633 nm)	BD Pharmingen™ Alexa Fluor^®^ 700	Human CD16	Dim	730/45	BD 560713

CD = Cluster of differentiation; PE = Phycoerythrin; PE-Cy7 = Phycoerythrin-Cyanine 7; PerCP-Cy 5.5 = Peridinin Chlorophyll Protein Complex-Cyanine 5.5.

## Data Availability

Data supporting the results and conclusions of this article will be provided on request moment by the corresponding author.

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
