# Peer review of "Dynamic Evaluation of Natural Killer Cells Subpopulations in COVID-19 Patients"

_ijms, 2022, doi:10.3390/ijms231911875_

Round 1

Reviewer 1 Report

Authors have aimed to evaluate the dynamic changes of the total NK cells and different NK subpopulations according to their differentiated expression of CD16/CD56 in SARS-CoV-2 infected patients. Study assessed the changes of NK cell subsets and their PD-1 expression through peripheral blood lymphocytes by flow cytometry, in different moments of disease evolution and also to correlate the NK subpopulation profile with the disease severity and outcome.

Following major comments needs to be carried out in their manuscript.

Abstract need significant improvisation. Especially concluding part of the work in the abstract.

I have observed many grammatical errors in throughout the manuscript. Authors are advised to check the errors and rectify it carefully. It was highly affecting the readability of the manuscript.

Lines 56-58, 59-68 etc: Syntax error. Reframe it.

Line 58: Authors are advised to explain/write about what kind of transcription factors involved?

Line 59: Natural Killer cells should be NK cells.

Line 66: First mention of MHC should be abbreviated.

Line 71: HLA means Human leukocyte antigen it was wrongly abbreviated as Major Histocompatibility Complex. Correct it.

Line 73: PBMC should be abbreviated.

The immune cells should be abbreviated in the first mentions.

Authors must concentrate on the formatting, and use of symbols, etc., There is no uniformity was observed in the manuscript.

Manicurist have many punctuation errors, check throughout the manuscript

Line 100-101: Reframe the sentence.

Lines 133-137: Syntax error. Reframe the sentence. It seems study…. studying; outcome …outcome.

Line 249: add comma after “Although, we did not find”

Materials and methods, results and discussion sections need English improvisation.

 Conclusion should be improvised with more detail and it will be useful to the readers for ease of understanding. Add some points about future prospects in conclusion section.

Throughout the manuscript need English and technical improvisations.

Reviewer 2 Report

Thank you very much for giving the opportunity to review this interesting work and appreciate much to all the authors. I have comments as below. 

1. Although the total number of participants is 53 at the admission point, it seems there is loss to follow-up or deaths so that only 49 in MS-5 and 20 in MS10 were followed up. The authors should explain the reason of such drop out cases as well as the timing of drop out cases and each number to each disease severity categories. It might effect on the analysis and results of the study. 

2. As the maximum follow-up period is 10 days, do the authors think it is enough follow-up period for critically/severe survivors cases for the assessment?

3.  According to demographic profile of participants in table 1, there is no information on presence or absence of comorbidity (underlying diseases), it can be the confounding on the result. Moreover, all severe/critical cases were old age group and death were occurred in this group only, how do the authors think of aging effect on immune system and the interpretation of the results. 

4. In figure 1, it is better to include the unit of measurement whether count or % in the column. Two figures do not have p value, if it was not calculates, please provide the reason. As the number of participants in each time point are different, the authors should provide the number of participants included for the analysis in each time point. 

5. How about the comparison of NK/NK subpopulation comparison by disease severity? 

6. The discussion is too long to draw the conclusion. The authors should consider to present/highlight the main facts and significant points of the study. 

Round 2

Reviewer 1 Report

Authors have addressed all the queriers. The manuscript was significantly improvised. But still I have found some minor corrections in the revised manuscript.

Line 23: Blood samples with EDTA

Line 24: you have written “day 5 day”, remove day after 5.

Line 35-39: Split the sentences in to two.

Line 93: Poor cytotoxic

Line 106-107: still it seems syntax error. Milush et collab. have particularly studied the CD16++CD56- subset of NK cells in particular,…. Check and remove the ‘in particular’

p value ‘p’ should be in italics in throughout the MS.

Author Response

Thank you for your appreciation.

We have made all the corrections except for the p-value in italic. From our previous experience, we noticed that the editors’ comment would be to change all italics letters in the upright letters for the final version. In addition, at this moment the article is overloaded with track-changing corrections so a process of correcting the p-values in italics would result in the loss of the previously entered information

Thank you for your appreciation.

We have made all the corrections except for the p-value in italic. From our previous experience, we noticed that the editors’ comment would be to change all italics letters in the upright letters for the final version. In addition, at this moment the article is overloaded with track-changing corrections so a process of correcting the p-values in italics would result in the loss of the previously entered information

Thank you for your appreciation.

We have made all the corrections except for the p-value in italic. From our previous experience, we noticed that the editors’ comment would be to change all italics letters in the upright letters for the final version. In addition, at this moment the article is overloaded with track-changing corrections so a process of correcting the p-values in italics would result in the loss of the previously entered information

Thank you for your appreciation.

We have made all the corrections except for the p-value in italic. From our previous experience, we noticed that the editors’ comment would be to change all italics letters in the upright letters for the final version. In addition, at this moment the article is overloaded with track-changing corrections so a process of correcting the p-values in italics would result in the loss of the previously entered information

Thank you for your appreciation.

We have made all the corrections except for the p-value in italic. From our previous experience, we noticed that the editors’ comment would be to change all italics letters in the upright letters for the final version. In addition, at this moment the article is overloaded with track-changing corrections so a process of correcting the p-values in italics would result in the loss of the previously entered information

Thank you for your appreciation.

We have made all the corrections except for the p-value in italic. From our previous experience, we noticed that the editors’ comment would be to change all italics letters in the upright letters for the final version. In addition, at this moment the article is overloaded with track-changing corrections so a process of correcting the p-values in italics would result in the loss of the previously entered information

Thank you for your appreciation.

We have made all the corrections except for the p-value in italic. From our previous experience, we noticed that the editors’ comment would be to change all italics letters in the upright letters for the final version. In addition, at this moment the article is overloaded with track-changing corrections so a process of correcting the p-values in italics would result in the loss of the previously entered information

Thank you for your appreciation.

We have made all the corrections except for the p-value in italic. From our previous experience, we noticed that the editors’ comment would be to change all italics letters in the upright letters for the final version. In addition, at this moment the article is overloaded with track-changing corrections so a process of correcting the p-values in italics would result in the loss of the previously entered information
